# *Candida glabrata*: Pathogenicity and Resistance Mechanisms for Adaptation and Survival

**DOI:** 10.3390/jof7080667

**Published:** 2021-08-17

**Authors:** Yahaya Hassan, Shu Yih Chew, Leslie Thian Lung Than

**Affiliations:** 1Department of Medical Laboratory Science, Faculty of Allied Health Sciences, Bayero University Kano, Kano 700241, Nigeria; hyahaya.mls@buk.edu.ng; 2Department of Medical Microbiology, Faculty of Medicine and Health Sciences, Universiti Putra Malaysia (UPM), Serdang 43400, Selangor, Malaysia; chewshuyih@upm.edu.my; 3Institute of Bioscience, Universiti Putra Malaysia (UPM), Serdang 43400, Selangor, Malaysia

**Keywords:** *Candida glabrata*, candidiasis, virulence factors, biofilm, antifungal drug resistance

## Abstract

*Candida glabrata* is a yeast of increasing medical relevance, particularly in critically ill patients. It is the second most isolated *Candida* species associated with invasive candidiasis (IC) behind *C. albicans*. The attributed higher incidence is primarily due to an increase in the acquired immunodeficiency syndrome (AIDS) population, cancer, and diabetic patients. The elderly population and the frequent use of indwelling medical devices are also predisposing factors. This work aimed to review various virulence factors that facilitate the survival of pathogenic *C. glabrata* in IC. The available published research articles related to the pathogenicity of *C*. *glabrata* were retrieved and reviewed from four credible databases, mainly Google Scholar, ScienceDirect, PubMed, and Scopus. The articles highlighted many virulence factors associated with pathogenicity in *C*. *glabrata*, including adherence to susceptible host surfaces, evading host defences, replicative ageing, and producing hydrolytic enzymes (e.g., phospholipases, proteases, and haemolysins). The factors facilitate infection initiation. Other virulent factors include iron regulation and genetic mutations. Accordingly, biofilm production, tolerance to high-stress environments, resistance to neutrophil killings, and development of resistance to antifungal drugs, notably to fluconazole and other azole derivatives, were reported. The review provided evident pathogenic mechanisms and antifungal resistance associated with *C. glabrata* in ensuring its sustenance and survival.

## 1. Introduction

Invasive candidiasis (IC) is a clinical condition that is not associated with a single *Candida* species. Each *Candida* species holds unique characteristics comparative to invasive potential, virulence, and antifungal susceptibility pattern [1]. It is an infection with many clinical manifestations that potentially affect any organs. Invasive candidiasis is associated with nosocomial bloodstream infections (BSIs) in tertiary health facilities worldwide [2]. *Candida* species also pose a significant threat to patients in the intensive care unit (ICU) with consequential mortality outcomes. They are the most commonly associated health care reported cases [3]. Major risk factors for *Candida* infections include prolonged usage of broad-spectrum antibiotics, immunocompromised state of the host, and the use of medical devices in surgery including catheters [3,4]. *Candida* species commonly cause invasive nosocomial infections in immunocompromised patients [5]. It accounts for 70–90% of all aggressive mycoses [6]. The increasing isolation of non-*albicans* species suggests increasing pathogenicity of these species with varying degrees of clinical symptoms [7].

*Candida glabrata* is an asexual, haploid yeast of the clade Nakaseomyces. It was initially named *Cryptococcus glabrata*. It then changed to *Torulopsis glabrata* in 1894, but the *Candida* genus was described in 1913 [8,9]. *Candida glabrata* is a successful pathogen colonising epithelial surfaces (mouth, gastrointestinal tract, vagina, skin, and present in stool) as healthy microbial flora with no age specificity [10]. *Candida glabrata* is commonly found in the environment, particularly on flowers, leaves, surfaces, water, and soil. It is the second most frequently isolated cause of candidiasis after *Candida albicans*. It accounts for approximately 15–25% of invasive clinical cases [8,11,12]. In fact, *C. glabrata* is the second most common species found in the United States and North-western Europe [1,11]. Increasing incidence of *C. glabrata* among *Candida* species as a cause of BSI in U.S. ICUs between 1989 and 1999 in a survey showed that *C. glabrata* ranked second to *C. albicans* accounting for 20% to 24% of all *Candida* BSIs [12]. Invasive candidiasis due to *C. glabrata* causes substantial morbidity and mortality of approximately 40–60%, perhaps due to the inherent low susceptibility of *C. glabrata* to the most commonly used azoles [3].

The usual route of *C. glabrata* to reach the bloodstream is through the breach of natural barriers, such as the use of catheters, trauma, or surgery [13]. However, disease susceptibility increases due to certain conditions such as AIDS and tuberculosis (TB), immunosuppressive use and cancer drugs, prolonged antibiotic therapy, and prolonged hospitalisation [14]. Increasing isolation frequency of *C. glabrata* is associated with old age, as reported by Zhang et al. [15]. Accordingly, *C. glabrata* was isolated more from patients in the age group >70 years than the other age groups (58.2% vs. 41.8%) out of 193 samples collected. A switch from normal flora to the pathogenic state may occur, leading to disease setting in, ranging from superficial (mucosal and skin) to systemic with an alarming mortality rate [16].

Virulence refers to the traits required for establishing a disease. However, strictly speaking, virulence factors have direct interaction and causing damage to the host cells [17]. Changes in the state of either the host or the microbe can affect the degree of virulence [18]. Many available factors facilitate the pathogenicity of *Candida* species. These include enzyme secretion, cellular adhesion, host defence evasion, and biofilm formation [7]. The infection thrives best in the presence of *Candida* species-specific virulence factors such as the presence of hyphae for invasion into host tissues [19]. *Candida albicans* filament exists in two distinct morphologies: hyphae and pseudohyphae. The expression of a specific gene set determines each morphology. The morphologies are critical as virulence factors occurring in most *Candida* species [20,21]. However, Galocha et al. [13] viewed that the pathogenicity of *C. glabrata* appears to be independent of the morphology of the yeast as this species is incapable of hyphae formation. Despite that, *C. glabrata* lacks several pathogenic attributes, critical in other *Candida* species, including polymorphic switching [22,23]; pathogenic relevance is alarming.

*Candida albicans* and *C. glabrata* show a significant difference in their mechanisms of virulence. *Candida glabrata* pathogenicity is associated with many virulence factors [24]. One of the most crucial factors is that it does not provoke a strong reaction by the host’s immune system. The treatment approach for *C. glabrata* infections is challenging due to the limited knowledge of its pathogenicity. The reduced antifungal drug susceptibility and the limited choices of effective antifungal agents are also challenging in treatment, as described by Yu et al. [25]. Other virulent factors include biofilm formation associated with adherence to host epithelial surfaces and hospital medical devices [7]. Despite the less destructive nature of *C. glabrata* in comparison to *C. albicans*, a high mortality rate associated with *C. glabrata* and rapidity of disease spread would argue otherwise [26]. *Candida glabrata* seems to have evolved a strategy based on secrecy, evasion, and persistence without causing severe damage in murine models [27]. Skrzypek et al. [28] also believed that *C. glabrata* exhibits a unique escape mechanism from the immune system and subsequently survives cellular engulfment and can resist antifungal treatment. This review summarises current information on the pathogenicity, virulence, and drug resistance mechanisms associated with *C*. *glabrata* (Figure 1).

## 2. *Candida glabrata* Virulence Factors

### 2.1. Enzyme Secretion

Secretion of hydrolytic enzymes is a significant determinant of pathogenicity in *C. albicans* and other non-*albicans* species. The enzymes protect against host defence reactions [29]. Phospholipases, proteinases, and haemolysins are powerful enzymes used by fungi to invade and infect susceptible hosts [30]. *Candida glabrata* secretes hydrolytic enzymes (e.g., phospholipases, proteases, and haemolysins) to destroy host tissues [19]. In addition to enzyme secretion, it is thought that host cell penetration occurs via endocytosis induction [13]. The study conducted by Nahas et al. [31] reported three gene families of phosphatases (*CgPMU1*-3) encoding phosphatase enzymes of different specificity. Accordingly, *CgPMU2* was identified as analogous to the *PHO5* gene found in *S. cerevisiae*. It serves as the phosphate-starvation inducible acid phosphatase gene. Almost all known candidal extracellular endopeptidases belong to the aspartic proteinase (Sap) class observed based on sequence analysis, proteolytic activity assay, and secretion of signal detection. *Candida glabrata* does not possess normal Sap genes in its genome [32]. In this context, *C. glabrata* is exceptional from this rule because the cell wall is associated with serine protease, Cwp1 (ORF: CBS138)—a gelatinolytic enzyme [24].

### 2.2. Adhesin Cell-Like Protein

*Candida* species initiate infection through adherence to host epithelial tissue and colonisation within the host [25]. *Candida* cell surface proteins involved in specific adherence to surfaces are described as adhesins, and they are critical in mediating biofilms’ formation [7]. *Candida glabrata* lacks yeast-to-hyphae switching, it grows only in the yeast form, contrary to the virulent switch of *C. albicans*. A significant virulence factor of *C. glabrata* is its ability to adhere firmly to many different substrates [3].

Cell surface adhesins in *Candida* species, particularly *C. glabrata* or *C. albicans*, have developed in large gene families [33]. The agglutinin-like sequence (Als) protein family and hyphae wall protein (Hwp1) in *C. albicans* are critical for the fungal adherence to host epithelial cells [34]. Unlike *C. albicans*, the main adhesins useful in *C. glabrata* originated from the epithelial adhesin (EPA) family. These adhesins facilitate *C. glabrata* attachment to host epithelial cells and assist in macrophage entry [25]. One such cluster includes a lectin-like *EPA* family. According to the mass spectrometric analysis obtained by De Groot et al. [35], 23 cell wall proteins were identified, including four novel adhesin-like proteins, Awp1/2/3/4 and Epa6. De Groot et al. [35] also reported that *C. glabrata* contains a unique, high number of genes encoding glycosylphosphatidylinositol (GPI) proteins from different clusters. Both (EPA and GPI) proteins are essential in adherence to human epithelial surfaces and biofilm formation. Cell wall components mediate interactions between *C. glabrata* and susceptible host, facilitating tissue adhesion and invasion. In addition, they are involved in biofilm formation, triggering the host immune response, and may confer resistance to antifungal drugs [36,37]. Notably, adhesin-like proteins in the cell wall depend on the stage of growth and the genetic background of the invading *C. glabrata*. Thus, the cells reflected alterations of adhesion capacity and cell surface hydrophobicity.

### 2.3. Biofilm Formation

Biofilms are considered biological communities formed by microorganisms with a high degree of organisation, structure, coordination, and functionality encased in a self-created extracellular matrix [36]. According to Kumar et al. [9], biofilm is a complex extracellular network of multi-layered microbial structures on biotic or abiotic surfaces shaped by microbe-microbe and organism–surface cooperation. The extracellular matrix defines the biofilm formed by all *Candida* species. In addition, the matrix contributes to pathogenicity by increasing drug tolerance and promoting immune evasion [38]. Biofilms formed by *Candida* species, including *C. parapsilosis*, *C. tropicalis*, *C. glabrata*, and *C. auris*, also associate with extracellular synthesis and high rich polysaccharides contents [38].

Both *C. albicans* and *C. glabrata* can form biofilms on abiotic substrates, especially medical devices including catheters and implanted materials [26,27]. Microbial biofilms can form in nature but also inside an infected host. Recently, there has been an increased relevance of microbial biofilms in human diseases, with an estimated 65% of all human infections being of biofilm aetiology [39]. Biofilm formation is another pathogenic mechanism observed in *C. albicans* with high biofilm mass, densely packed with pseudohyphae. However, *C. glabrata* produces sparse biofilm (less weight) with yeast cells. Thus, it is an essential pathogenic mechanism for its survival [40] (Figure 2).

Candidiasis associated with biofilm production has clinical implications. The formation of biofilm on medical devices can cause device failure. In addition, it can serve as a point source for further infections [41]. Fungal biofilms show properties different from planktonic (free-living) populations, including a higher antifungal resistance level [39]. The resistance development due to biofilm is complex and multifactorial; among the assumed mechanisms are (i) the elevated cellular density within the biofilm, (ii) the exopolymeric protective effect of the biofilm, (iii) differential upregulation of genes linked to resistance and those encoding efflux pumps, and (iv) the presence of a subpopulation within the biofilm community.

The emergence of echinocandins and liposomal formulations of amphotericin B drugs show increasing efficacy against fungal biofilms [36,39]. Recent evidence indicates that most IC caused by *C. glabrata* is associated with biofilm growth [42]. *Candida glabrata* biofilms show antifungal resistance characterised by a compact, dense structure of yeast cells. The cells become nested in an extracellular matrix composed of high proteins and carbohydrates β-1,3 glucan contents [9]. Several genes are associated with biofilm formation in *C. glabrata*. For example, the *EPA6* gene encodes adhesin regulated by multiple factors, including the CgYak1p kinase, subtelomeric silencing, chromatin remodelling Swi/Snf complex components, and the transcription factor CgCst6, which plays an important role.

Moreover, adhesins, cell wall proteins, and RNA polymerase II mediator complex subunits, including Epa3, Epa7, Epa12, Awp4–6, Pwp1, Pwp3, Med12, Med13, and Med15, results in biofilm formation [43]. According to da Silva Dantas et al. [44], low-level colonisation of epithelial surfaces may create a mature surface biofilm. Nevertheless, it is unclear how the biofilm structure formed by *Candida* affects mucosal surface infection and host immunity. However, such mature biofilms formed with dense biomass would severely challenge the cellular immune system in containing and clearing them from the host system. According to Jeffery-Smith et al. [45], *C. auris* biofilms demonstrated higher biomass than *C. glabrata* and reduced biomass compared with *C. albicans*. Resistance to drug sequestration in the biofilm matrix also reduces drug efficacy. It lowers the exposure of *C. glabrata* to the drug, facilitating the selection of acquired resistance [43]. Al-Dhaheri and Douglas [46] found that the presence of persister cells in biofilms is mainly responsible for biofilm resistance. Accordingly, *C. krusei* and *C. parapsilosis* appear to possess persister cells that may become tolerant to drugs. In contrast, biofilms of *C. glabrata* and *C. tropicalis* do not possess such persister cells [13,43].

### 2.4. Presence of a Stable Cell Wall

The fungal cell wall is the primary contact site for host-pathogen interaction [47]. The fungal cell wall consists of complex biomolecule structures made up of polysaccharides, proteins, and lipids. The composition is dynamic, responding to changes in the local environment [25,48]. *Candida* cell wall consists of an inner layer of polysaccharides (chitin, 1,3-β-glucans, and 1,6-β-glucans). An outer layer of proteins glycosylated with mannan constitutes the pathogen-associated molecular patterns (PAMPs). The PAMPs are recognised by specific innate immune receptors known as pathogen recognition receptors (PRRs) [20]. The cell wall is dynamic and necessary to maintain the osmotic pressure exertion and morphology during vegetative growth. Other environmentally induced developmental changes such as sporulation, sexual reproduction, or pseudohyphae growth are often necessary for survival and growth. The fungal cell wall comprises three significant polysaccharides: glucans, mannoproteins, and chitin [49]. Moreover, the findings of Srivastava et al. [50] showed that cysteine abundance is common in fungal extracellular membranes (CFEM) domain-harbouring cell wall structural protein, CgCcw14, and a putative haemolysin, CgMam3. They are vital for the maintenance of intracellular iron content, adherence to epithelial cells, and virulence.

During fungal growth, the cell wall expansion causes permanent remodelling of the polysaccharide network, consisting of mannans, β-glucans, and chitin. Chitin is a homopolymer of β-1,4-N-acetylglucosamine (GlcNAc). Chitin is essential for fungal biological functions, including cell division, septa formation, hyphal growth, and virulence [47]. The chitin synthases enzyme carries out chitin synthesis in *C. glabrata*. Deregulation of chitin biosynthesis is a potential mechanism of virulence and resistance to antifungal therapy—the presence of drugs, such as echinocandin, results in the corresponding increase in chitin synthesis. The chitin maintains the cell wall’s structural integrity, as chitin replaces β-1,3-glucan. High chitin content restricts the penetration of the drug through the cell wall [51]. *Candida glabrata* presents strange features related to cell wall organisation, such as overexpression of genes encoding adhesion-like GPI-anchored proteins or the implication of GPI-anchored aspartyl proteases (yapsins) in the infection process. These features indicate key virulence factors, with multiple roles in the high tolerance to azole drugs, adhesion to susceptible host cells, or survival inside macrophages [52].

Genetic mutations confer susceptibility to patients against *Candida* species [20]. *Candida glabrata* has well-characterised genes, including *ACE2* (*CgACE2*), a transcription factor that serves as a negative regulator of virulence. It was studied in an invasive infection of an immunocompromised mice model. The evolved (Evo) strain is another hyper-virulent *C. glabrata* strain with a single nucleotide mutation in the chitin synthase gene *CHS2*. Both mutants have enhanced virulence. Moreover, they stimulate inflammatory response factors, such as tumour necrosis factor-alpha (TNF-α) and interleukin-6 (IL-6). Thus, the *ace2* mutant and Evo strain exhibit a clumpy pseudohypha-like structure [25]. Other strains with enhanced virulence characters include a strain with the *PDR1* gain-of-function mutation, a strain with mitochondrial dysfunction, and the *anp1* and *mnn2* glycosylation mutants [25].

### 2.5. Novel Hybrid Iron Regulation and Acquisition Strategies

*Candida glabrata* requires iron as an essential micronutrient for its growth during infection. Thus, it is necessary to strategize the mechanism for its acquisition for disease establishment [53]. Among the known iron uptake mechanisms in fungi are siderophore-interceded uptake of Fe^3+^, reductive iron procurement, and haemoglobin/haem uptake. All these frameworks are operational in *C. glabrata* except for the receptor-interceded haem uptake [9]. The underscore tight regulation of all processes involving iron in the organism, including uptake, distribution, utilisation, and storage. *Candida glabrata* has high-affinity iron uptake mechanisms as critical virulence determinants.

Hosts’ fundamental approach uses ‘nutritional immunity’ to limit the iron required by invading pathogenic microorganisms, similar to in humans, available iron seized by various carriers and storage proteins, including haemoglobin, transferrin, and ferritin. They virtually deprive the available iron system, leaving no option for invading organisms. It, thus, exploits other iron source mechanisms (reductive, non-reductive, and haemoglobin-bound iron acquisition and degradation) [50]. Iron is usually incorporated into haem or bound iron-sulphur, acting as a cofactor in many vital processes. These processes include the tricarboxylic acid cycle (TCA), DNA replication, mitochondrial respiration, and detoxification of reactive oxygen species (ROS) [54]. Iron effectively works due to its redox potentiality to switch between the two states as ferric iron (Fe^3+^) and ferrous iron (Fe^2+^). Both ionic states have different effects on pathogenic microorganisms. For instance, Fe^3+^ is poorly soluble in alkaline conditions, and Fe^2+^ becomes toxic by promoting ROS production via the Fenton reaction [55]. According to the findings of Srivastava et al. [50] that the high-affinity reductive iron uptake system is necessary for metabolism in the presence of alternate carbon sources and for growth under both in vitro and in vivo iron-limiting conditions. The phenotypic, biochemical, and molecular analyses of 13 *C. glabrata* strains deleted for proteins (Cth1, Cth2, and common in fungal extracellular membranes (CFEM) domain-containing structural proteins CgCcw14, CgMam3, and putative haemolysin) confirmed that these proteins are potentially implicated in iron metabolism.

While *Saccharomyces cerevisiae* is a non-pathogenic yeast belonging to whole-genome duplication clade (WGD), having significant similarities with pathogenic *C. glabrata* [3], it is poorly understood whether the different pathogenic clades, including CTG, may use common infection strategies or lineage-specific mechanisms or both combinations for pathogenicity [3,53]. *C. glabrata* combines the iron regulation network properties of both pathogenic and non-pathogenic fungi (*S. cerevisiae*). *Candida glabrata*, such as *S. cerevisiae*, uses the *Aft1* gene as the primary positive regulator during the sub-optimal iron condition. At the same time, *Cth2* degrades mRNAs encoding iron-requiring enzymes. However, it contrasts with *S. cerevisiae* in that it requires *Sef1* ortholog for total growth under iron-limited conditions. The iron homeostasis mechanisms in *C. glabrata* is still unknown. *Candida glabrata* showed host-specific iron acquisition mechanisms by utilising siderophores and haemoglobin as a source of iron and haemolysin. It also uses cell wall structural protein to maintain iron homoeostasis [50].

### 2.6. Adaptation to Various Environmental Conditions

Yeast cells within their natural habitat make many metabolic adjustments in response to changes in extracellular environmental nutrients. Such changes result in gene expression, which are either upregulated or downregulated depending on the environmental requirements [56]. Adaptation of gene expression through transcription regulation is a significant mechanism in fungal response to rapidly changing environmental conditions [57]. The response was first described in *Saccharomyces cerevisiae* and is referred to as general stress response or environmental stress response (ESR). Genome-wide environmental stress response (ESR) expression profile of *C. glabrata* is coordinated by Msn2 which is the main transcriptional response activator. Transcription factors Msn2 and Msn4 are crucial for resistance to various stresses in *C. glabrata* [58]. Activation of Msn2 and Msn4 in the cells causes their rapid accumulation in the nucleus and recruitment to chromatin. Msn2 has separate functional domains for nuclear import (nuclear localization signal, NLS), nuclear export (nuclear export signal, NES), and DNA binding. The stress conditions including disturbed cellular integrity, osmostress, elevated temperature, and the presence of antifungal drug resistance are commonly observed in clinical isolates [22].

*Candida* species can quickly adapt to host environmental changes as commensal pathogens even under nutrients bioavailability restriction [13,59]. *Candida* species use different nutrients available in the vast host niche. The *Candida* pathogens possess a high degree of metabolic flexibility because of the adaptive metabolic mechanisms necessary for significant nutrient acquisition [60]. Fungal pathogens require the adaptation to different host immune defence mechanisms and environmental stresses.

Environmental parameters including temperature, pH, serum, and CO_2_ are associated with several steps during host invasion and optimal growth of *Candida* species [13]. *Candida* species can withstand a wide range of temperatures and pH as virulence factors [61]. *Candida glabrata* grows optimally at 37 °C and, therefore, thrives best in the human host and can grow at 42 °C under heat-stressed conditions [18,57]. Temperature variability affects gene expression and can result in induction or repression of genes encoding functions linked to virulence [62]. A study conducted on the virulence of *C. glabrata* on the *Galleria mellonella* model indicated that *G*. *mellonella* only became susceptible to infection at 37 °C. Thus, this suggested that some essential genes for *C. glabrata* virulence are switched on only at 37 °C [62].

Flexibility in carbon metabolism is critical for the survival, propagation, and pathogenicity of many human fungal pathogens [60]. According to the findings of Chew et al. [63], the growth of *C. glabrata* in the presence of acetate, lactate, ethanol, or oleate reduces the growth in both the planktonic and biofilm states. The use of glucose as a carbon source, on the contrary, showed significant growth in both states. Moreover, the findings reported the necessity of isocitrate lyase (*ICL1*), the glyoxylate cycle gene for acetate utilisation, ethanol, and oleic acid, and partly required for the utilisation of lactate in *C. glabrata*.

The mechanism of acid stress tolerance in *C. glabrata* has not been extensively investigated. The low pH of *C. glabrata* cultures during pyruvate production causes a slow or total halt in growth due to acid accumulation [64,65]. Contrary to the view of Yan et al. [66] that overexpression of the transcription factor CgCrz1p enhances viability, cellular biomass, and pyruvate yields at a low pH. Accordingly, CgCrz1p might serve a significant role in the integrity and fluidity based on the analysis of plasma membrane lipid composition. Thus, it enhanced the pumping of protons in acidic environments. *Candida glabrata ASG1* (CgASG1, CAGL0G08844g) deletion resulted in increased tolerance to salt stress [58]. Active pH modulation is one likely fungal approach to change the pH of the phagosome. *Candida glabrata* makes its extracellular environment alkaline when grown on amino acids as the sole carbon source in vitro. Mutant *C. glabrata* that lacks fungal mannosyltransferases resulted in strictly reduced alkalinisation in vitro. The condition subjects *C. glabrata* to acidified phagosomal activity [21]. Proteomic analysis of the pH response showed that *C. glabrata* observes low pH as less stressful than high pH [58]. The low acidic environment of the vaginal tract (pH ~ 4–4.5) contributes to the increased resilience to azoles against *C. glabrata* and *C. albicans.* Thus, this demonstrates the decreased efficacy of azole drugs in vitro at acidic pH [67].

During phagocytosis, the internalised microbes become lysed in lysosomes—a specialised compartment in which oxidative and non-oxidative mechanisms kill and degrade the internalised microbes [21]. *Candida glabrata* lacks hyphal formation and phagosomal extrusions to escape the phagocytic cells attack contrary to *C. albicans* [68,69]. In *Cryptococcus neoformans*, the produced capsules inhibit phagocytosis by macrophages and prevent the killings of the already internalised cells [70]. The less aggressive mechanism helps in an autophagy process by mobilising its intracellular resources for metabolism and survival during prolonged starvation [68,69] Evidence suggests that growth in the presence of alternative carbon sources affects the phagocytosis of *Candida* species. *C. glabrata* has high-stress resistance. Perhaps its enhanced sustenance during starvation allows it to survive and replicate inside the immune system cells (macrophages). The *C. glabrata* are engulfed during bloodstream circulation [13,18]. Chew et al. [71] revealed that the *ICL1* gene helps promote the growth and prolonged survival of *C. glabrata* during macrophage engulfment. Thus, *C. glabrata* shows a unique immune system evasion mechanism and survives after cellular engulfment despite the antifungal presence. Perhaps through concealment within intracellular niches [21,28]. Lactate-grown *C. glabrata* cells, for example, resist killing by macrophages and have developed distinct tactics for intracellular survival killing and escaping phagocytosis [41]. Following extended division, the macrophages rupture, and yeast cells escape and disseminate into the blood system for further spread [13] (Figure 3).

Successful clearance of pathogens depends on phagocytes’ rapid actions of the innate immune system, such as macrophages, dendritic cells, and neutrophils [21]. The primary factor aiding the persistence of *C. glabrata* is its less aggressive nature to stimulate the strong reaction of the host immune system [24]. Because of the low host cell damage, *C. glabrata* cells elicit a cytokine profile significantly different from that of *C. albicans*. Consequently, *C. glabrata* is associated with mononuclear cell proliferation (macrophages). In contrast, neutrophil emergence becomes typical of *C. albicans* [8]. Despite the medical importance of *C. glabrata,* it is less lethal because it provokes a low inflammatory immune response. The systemic mouse infection models indicated that even at high inocula doses of intravenous infection [21]. Furthermore, the upregulation of Trx1p as a stress-response protein exerts defences to *C. glabrata* against oxidative stress [72]. Considering the role of dimorphism as a factor for pathogenicity in some *Candida* species, *C*. *glabrata* is exceptional; it does not germinate into hyphae yet is virulent [73].

### 2.7. Replicative Ageing

*Candida glabrata* as occur in *S. cerevisiae*, *C*. *albicans*, and *C. neoformans* show a replicative ageing, a process where original mother cells progressively age, producing asymmetric mitotic divisions resulting in phenotypically distinct daughter cells [16]. It can also contribute to the microevolution of pathogens in a specific host [74]. A mother cell can only produce a specific number of buds during mitotic division. The total number of buds that a mother cell produces before the division ceases and dies is the designated replicative life span (RLS). Each cycle of bud formation by a mother cell represents one generation [75]. Several studies showed that replicative ageing in many fungal pathogens leads to significant changes that affect the fungal resistance to phagocytic clearance and antifungal therapy [75]. The phenotypic changes in the daughter cells due to ageing are not genetically inherited. The old cells only emerge because of neutrophil pressure in the environment that favour the killing of young fungal cells and the promotion of the persistence of old cells [75]. Thus, for the pathogen, this form of adaptation is advantageous, as it avoids the risk of random permanent mutations and instead assures that all adaptive changes are easily reversed in the daughter cells that are borne from asymmetric budding. Aged cells exhibit different lipid composition that leads to the emergence of azole resistance. The replicative age allows the transition from commensalism to a pathogenic state. The intimate association between *C. glabrata* and a mammalian host may result in resilience and high-stress tolerance. The host becomes vulnerable to invasive diseases during neutropenic or immunocompromised states [74].

*Candida glabrata* can shift from a commensal to pathogenic state due to the pressure of neutrophils. Bouklas et al. [74] reported a controlled depletion in studies of *C. glabrata* in the murine models. The findings indicated that ageing leads to remodelling of the cell wall and that neutrophils selection controls generational distribution within the *C. glabrata* population. The in vivo study by Bhattacharya et al. [76] viewed that the neutrophils cells in the host selectively kill younger cells, leaving the old yeast cells to accumulate. Perhaps, the ageing *C. glabrata* mother cells’ large cell sizes and thicker cell walls contribute to their better resistance to neutrophil killings than the young daughter cells.

## 3. Drug-Resistance Mechanisms of *Candida glabrata*

The emergence of antifungal resistance becomes a problem in clinical medicine, significantly when associated with *Candida* species. Knowledge of *C. glabrata* infection symptoms is essential because *Candida* species commonly share indices of suspicion of the disease. *C. glabrata* among the non-*albicans Candida* species can acquire drug resistance. Moreover, it can develop secondary resistance to other available antifungal classes, resulting in poor treatment outcomes. It is a well-known fact that both *C*. *krusei* and some *C. glabrata* have intrinsic resistance to fluconazole. In such a situation, proper diagnosis is essential to justify appropriate treatment [77].

The incidence of candidemia caused by fluconazole-resistant strains and derivatives is high [59]. Azole drugs are among the four classes of antifungals commonly used in clinical practice to treat cancer, AIDS, patients on chemotherapy, and bone marrow transplant patients with fungal infections [78]. The most prevalent *Candida* species, *C*. *albicans* and *C. glabrata* differ significantly in response to antifungal therapy [79]. Fluconazole is extensively prescribed and administered because of its availability for oral administration, has low toxicity, and is less expensive. However, the extensive use of fluconazole has led to the increasing emergence of resistant isolates [80,81].

*Candida glabrata* infections are complicated to treat due to their inherent resistance to antifungals, especially against azoles [41]. Sardi et al. [42] viewed that *C. glabrata* has intrinsic antifungal resistance, especially to fluconazole. Arendrup and Patterson [43] argued that *C. glabrata* developed acquired resistance to antifungal drugs through prolonged exposure. Moreover, Jensen et al. [82] supported the view that prolonged administration of antifungal drugs for treatment and prevention is the primary cause of the emergence of resistant strains. The frequency and relatively high mortality rates of these infections are generally associated with pathogenic yeast capacity to efficiently develop multiple drug resistance (MDR).

Moreover, *C. glabrata* shows multi-drug-resistant capacity at an alarming rate. The genomes of *C. glabrata* can accumulate gene mutations that result in phenotypic resistance to antifungals after exposure to multiple drugs [83]. For example, mutations in the *MSH2* gene, encoding a DNA mismatch repair protein, occur in *C. glabrata.* Its effects have been found in clinical isolates to facilitate the selection of resistance to azoles, echinocandins, and polyenes in vitro [1]. On a general note, the published in vitro data have shown that deoxycholate amphotericin B (dAmB) and echinocandins such as caspofungin or micafungin demonstrated high activity against *C. albicans* and *C. glabrata* growing in biofilms settings [84].

### 3.1. Types of Drug Resistance Mechanisms

#### 3.1.1. Azole Resistance

Azole drugs play a critical role in clinical practice, especially fluconazole, clotrimazole, and imidazoles [78]. Fluconazole is the frontline drug used for prophylaxis and treatment of many fungal infections [85]. The disease candidiasis has predisposing factors including organ and bone marrow transplant, prolonged chemotherapy, and AIDS [78]. The reported ability of *C. glabrata* to show resistance to fluconazole in clinical isolates indicates the need to improve the diagnostic approach. In addition, it promotes new antifungal therapy for easy management of such cases [35].

*Candida glabrata* possesses numerous resistance mechanisms to fluconazole, including fluctuation of gene regulation, genetic mutations, and cross-resistance among azole derivatives [86]. Yoo et al. [81] described the primary tools of azole resistance associated with *Candida* species, including mutations in the *ERG11* gene and the proliferation of copy number of azole targets. Other mechanisms include blockage of the ergosterol biosynthesis pathway. Mutations in *ERG11* and *PDR1* can mediate azole resistance; daughter cells will inherit the mutations and persist [76]. Over-expression of genes coding some adenosine triphosphate (ATP)-binding cassette is fluconazole resistance mechanism of *C. glabrata* as observed in Iranian isolates. Resistance mechanisms are also associated with significant facilitator superfamily efflux pumps, leading to the increasing efflux of azole drugs. Although numerous possible tools have been reported previously, the exact resistance mechanism is not entirely clear on azole resistance. Approximately more than 140 alterations in the *ERG11* target gene have been described. Some alterations are exclusively found in azole-resistant isolates, whereas some are obtained in susceptible isolates [43].

The mechanism of action of azole is to target the cytochrome P450 enzyme sterol 14α-demethylase. The enzyme converts lanosterol to ergosterol as an essential structural component of the fungal cell membrane [87]. According to the findings of Gohar et al. [86] and Farahyar et al. [88], the ATP-binding cassette transporters of drug efflux is mediated primarily by *Candida glabrata* sensitivity to 4 Nitroquinoline N-oxide (*CgSNQ2*) and *Candida glabrata Candida* drug resistance 1 and 2 (*CgCDR1* and *CgCDR2*) genes. More specifically, the free nitrogen atom of the azole ring binds an iron atom within the enzyme haem group. Thus, it prevents oxygen activation and causes demethylation of lanosterol that inhibits the ergosterol biosynthesis process. The inhibition is toxic methylated sterols accumulated in the fungal cellular membrane, and cell growth is arrested [89].

According to Poláková et al. [90], the genetic instability results in segmental duplications, chromosomal rearrangements, and extra chromosomes occurring in *C. glabrata* at high frequency. Several genes on chromosomes (ChrE_L_ and ChrF_L_) potentially mediate interactions between *C. glabrata* and the susceptible host. Duplicated segments of ChrFL encode a transporter of the ATP-binding cassette family (CAGL0F01419g) that is very similar to *S. cerevisiae AUS1*. The small chromosome F encodes an ortholog of *S. cerevisiae* ABC transporter *PDR5* (CAGL0F02717g) known in *C. glabrata* as *PDH1*. Torres et al. [91] viewed that aneuploidy causes a transcriptional response that results in gene expression in chromosomes. Aneuploidy gain of small chromosome segment on the left arm of chromosome F that encodes ABC transporter *AUS1* and *PDH1* is also observed in *C. glabrata*-resistant isolates [90]. Duplications increase the level of drug resistance, as ABC transporters are implicated in pleiotropic drug resistance.

The major facilitator superfamily (MFS) is a membrane transporter that helps in accomplishing the active efflux of azole. MFS transporters facilitate enhanced fluconazole efflux particularly in ageing *C. glabrata* cells [76].

Mitochondrial DNA deficiency is another mechanism used by *C. glabrata* for azole resistance through the upregulation of ATP-binding cassette (ABC) transporter genes. The upregulation of these transporters is associated with gain-of-function (GOF) mutations in the transcriptional regulator encoded by *CgPDR1*. Cells with mitochondrial DNA deficiency are called ‘petite mutants’ [92]. Ferrari et al. [93] reported two *C. glabrata* isolates (BPY40 and BPY41) obtained from the same patient on different occasions. The former was azole sensitive, while the latter was azole-resistant. Upon testing, BPY41 showed mitochondrial dysfunction compared to BPY40. The virulent analyses, based on mortality and fungal tissue burden in both systemic and vaginal murine infection models, suggested higher virulence of BPY41 than BPY40. Then, oxido-reductive metabolism and the stress response were also observed in the BPY41 isolate. Based on the microarray analyses, some genes responsible for cell wall remodelling were upregulated in BPY41 compared to BPY40. These pieces of evidence suggested that virulence and resistance to azole were linked to mitochondrial dysfunction in BPY41.

For instance, Nedret Koc et al. [94] reported three *C. glabrata* isolates with MICs of ≥8 µg ml^−1^. One *C. glabrata* isolate showed MICs of ≥1 µg mL^−1^ and two *C. glabrata* isolates showed to have MICs of ≥1 µg mL^−1^. Song et al. [95] reported from South Korea that two of the five *C. glabrata* isolates tested were resistant to fluconazole. The five isolates were resistant to itraconazole. Similarly, all the isolates were resistant to itraconazole. In a similar resistance pattern, Premamalini et al.’s [96] study conducted in Chennai, India, indicated that two (66.7%) *C. glabrata* isolates were resistant to fluconazole and itraconazole. However, the isolates were susceptible to voriconazole. A study conducted in China reported that 12.2% of *C. glabrata* are resistant to fluconazole. The result also showed a 17.8% susceptibility of the standard strains to voriconazole [97]. These findings agreed with the view of Larkin et al. [98] that newer azoles, such as posaconazole and voriconazole, replaced fluconazole in prophylaxis routines and have raised similar concerns about resistance and drug-drug interactions as observed in azoles. According to the clinical laboratory standard institute (CLSI) M27-A2, the interpretative breakpoints for fluconazole for in vitro testing of *Candida* species using broth microdilution method is ≤8 µg mL^−1^ for susceptible (S), 16–32 µg ml^−1^ for susceptible-dose dependent (S-DD), and ≥64 µg mL^−1^ for resistant (R) [99].

#### 3.1.2. Echinocandins Resistance

Echinocandins are recommended as first-line therapy for non-neutropenic patients associated with *C. albicans* and *C. glabrata* in suspected severe IC conditions [100]. Echinocandins were launched in the early 2000s and became the first-line treatment following the emergence of *C. glabrata* with reduced susceptibility to fluconazole Colombo et al. [101]. The emergence of resistant *C. glabrata* usually correlates with high azole and frequent echinocandin usage in hospitals or specific hospital wards [1]. *Candida* species strains resistant to first-line antifungals (such as fluconazole and echinocandins) are increasingly documented. Echinocandins exist according to the international guidelines in three available types (caspofungin, anidulafungin, and micafungin). They differ based on the route of metabolism, half-life, and safety [102].

The echinocandins drugs act by inhibiting β-d-glucan synthase, an enzyme necessary for cell wall synthesis. They have excellent fungicidal activity against most *Candida* species [1]. The prevalence of echinocandin resistance among *Candida* strains remains low, with around 4% observed in *C. glabrata* and less than 1% in *C. albicans* [103]. Intrinsic resistance to echinocandins is rare and primarily only observed in *Candida parapsilosis* [103]. However, Echinocandins resistance is an emerging scourge in *Candida* species, particularly *C. glabrata* [1]. In infections associated with *C. glabrata* or *C. krusei*, echinocandins offered preference over azoles. One major setback of echinocandins oral administration is the inadequate bioavailability, and therefore, it is often given intravenously [104]. The frequent prescription and usage of echinocandins result in resistance development by decreasing the targeted *Candida* species susceptibility [102]. Despite the indication of echinocandin efficacy as antifungal prophylaxis, some concerns increasing patient exposure will lead to echinocandin resistance, particularly in *C. glabrata.* The resistance acquired through mutation (amino acid changes) of the critical regions in *FKS1* and *FKS2* genes encode β-1,3-d-glucan synthase, the potential target enzyme echinocandins [51,103].

Mutations frequently occur in *FKS2* relative to *FKS1* [101]. Shields et al. [105] reported low echinocandin resistance, around 4% for *C. glabrata* and less than 1% for *C. albicans*. However, Sasso et al. [103] reported increasing isolation of echinocandin *C. glabrata*-resistant strains, significantly associated with FKS1 and FKS2 gene mutations. Mutations in two hotspot regions (HS1 and HS2) of these genes have been recognised as the primary mechanism for echinocandin resistance [1]. Based on Aslani et al. [106] findings in the study conducted in Iran on echinocandins, 27.8% of the *Candida* isolates showed resistance to caspofungin. All isolates were highly susceptible to anidulafungin except *C. glabrata* with 10% resistance. The SENTRY surveillance program between 2006 and 2010 reported 11% echinocandin and fluconazole resistance among *C. glabrata* (i.e., MDR) [101].

#### 3.1.3. Polyenes Resistance

Amphotericin B (AmB) is a fungicidal polyene and has shown promising activity against many *Candida* species. It is used in the pharmacotherapy of life-threatening fungal infections [107]. Despite these therapeutic advantages, AmB has serious toxicity limitations on the human host cells. This is because both human and fungal cells’ biomembranes are the primary targets of the AmB. Thus, impairing the physiological processes that take place in the membranes, particularly adenocarcinoma cells [108]. Most of the published practices with AmB for the treatment of IC reported the deoxycholate preparation of the AmB (AmB-d). Two lipid formulations of AmB (LFAmB) have also been developed. They are generally available as an AmB lipid complex (ABLC) and liposomal AmB. The formulations possess the same spectrum of activity as AmB-d against *Candida* species. However, they differ based on the daily dosing regimens and toxicity profiles. Amphotericin formulations are the best therapeutic option, mainly in catheter-related bloodstream infections in neutropenic patients [101].

The mechanism of action of AmB is to bind to ergosterol in the plasma membrane resulting in the leakage of cytoplasmic materials and cellular destruction [51,98]. The resistance to AmB is not commonly observed in *Candida* species [11]. Some studies have linked mutations in *ERG2*, *ERG3*, *ERG5*, *ERG6*, and *ERG11* genes with the depletion of ergosterol as a significant cause of AmB resistance [109]. Tay et al. [110] reported that *C. glabrata* isolates demonstrated similar MIC_50_ (0.25 μg/mL) against AmB for biofilm and planktonic cells. The findings attributed lower resistance of *C. glabrata* with biofilms against amphotericin and not about the low biofilm content of the isolates tested. The findings agreed with the study reported by Al-Dhaheri and Douglas [46] that ‘persister’ populations were observed in biofilms of *C. albicans*, *C. krusei,* and *C. parapsilosis* after exposure to amphotericin. Such a ‘persister’ population was absent from the biofilms of *C. glabrata*. In contrast, Rodrigues et al. [107] viewed that *C. glabrata* can produce biofilms in the presence of AmB therapeutic concentrations due to the high concentrations of carbohydrate and β-1,3 glucan on the biofilm matrices. This underlines the capacity of *Candida* cells to rapidly adjust to external aggressions. Thus, this suggests why patients undergoing AmB therapy may still manifest resilient *Candida* infections. According to the findings of Bhattacharya et al. [76], replicative ageing in *C. glabrata* causes higher tolerance to killings by AmB and micafungin due to the higher transcription of glucan synthase gene, *FKS1*. The study of Aslani et al. [111] reported that 39% of yeast strains from cancer patients showed higher MIC values to AmB, with MIC_90_ values of 4 μg/mL. According to the findings reported by Norimatsu et al. [4], the liposomal AmB (3 mg/kg/day) showed better activity than micafungin in the treatment of both *C. glabrata* and *C. parapsilosis* bloodstream infections in the case of an 80-year-old woman.

## 4. Conclusions and Way Forward

The alteration of host interaction factors facilitates opportunistic harmless commensal *Candida* species to become potentially life-threatening human pathogens. Management of candidiasis involves the identification and control of host predisposing factors to infection. Moreover, *Candida* species possess many virulence factors, including secretion of hydrolytic enzymes host cell wall adherence, and biofilm formation. The factors help to increase their persistence and survival. *Candida glabrata* further adopt strategies of evading the action of immune cells. *Candida glabrata* cells become engulfed in the macrophages. They quickly adopt other mechanisms of survival by undergoing massive division and release upon rupture. Thus, it is essential to study the functions of those genes associated with that mechanism because of increasing mutation occurrence. Constant surveillance of antifungal susceptibilities in clinical isolates of *C. glabrata* at the national and international levels is necessary to control the spread of resistance. The survey will provide practical strategies for the prophylaxis and treatment of human infections associated with *C. glabrata.* Moreover, acquired and intrinsic resistance to fluconazole and rapid resistance development to a few available antifungal drugs are of significant clinical concern. These issues prompt the urgent development of a better diagnostic approach to detect and identify *C. glabrata* for effective and timely treatment and management. Diagnostic development of methods with robust sensitivity, specificity, and short turn-around time could significantly assist to better manage patients with candidiasis caused by *C*. *glabrata*.

## Figures and Tables

**Figure 1 jof-07-00667-f001:**
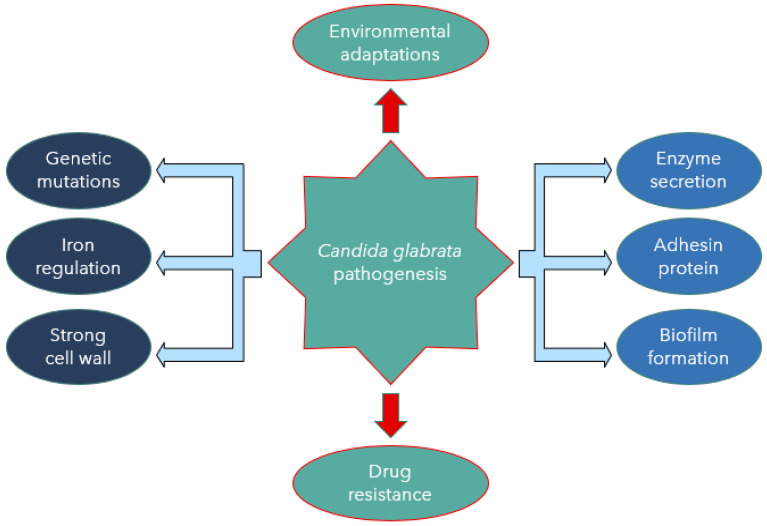
*Candida glabrata* pathogenesis mediated by virulence factors.

**Figure 2 jof-07-00667-f002:**
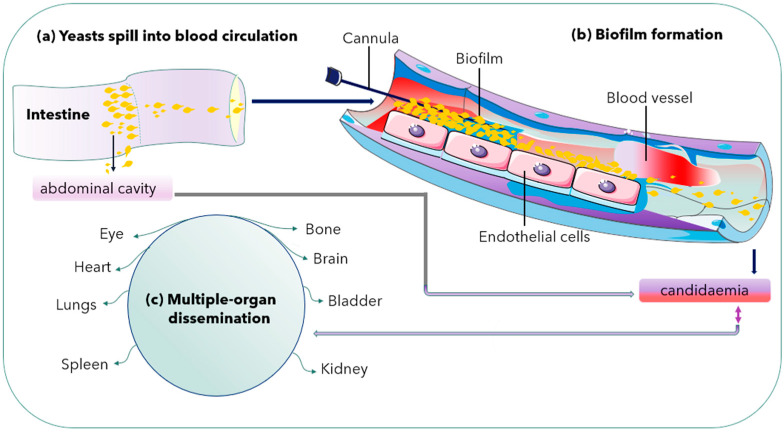
Biofilm formation in a blood vessel and dissemination into multiple organs. Double arrow shows either way dissemination of *C. glabrata* cells.

**Figure 3 jof-07-00667-f003:**
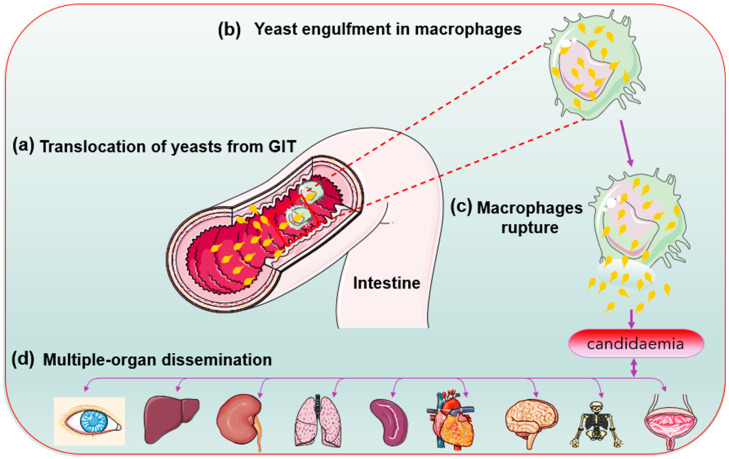
*Candida glabrata* cells (yellow) replication inside the macrophage cells before organ dissemination.

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
