# Peer review of "Candida glabrata: Pathogenicity and Resistance Mechanisms for Adaptation and Survival"

_jof, 2021, doi:10.3390/jof7080667_

Round 1
Reviewer 1 Report
General impression
The authors have submitted a review of C. glabrata survival mechanisms inside the human host. While the outline of the review and the major conclusions are appropriate, the writing leaves much to be desired. Citations are not universally accurate, and the grammar and organization of the text are sometimes confusing. It is not clear who the intended audience is (physicians looking for treatment recommendations? Researchers looking for a brief introduction into the field?) or why the review is necessary (no major new findings are discussed). The conclusions are not clear: the authors are making a rather obvious point for maintaining resistance surveys, but it is not clear why better diagnostic approaches are needed as diagnostic approaches and their shortcomings are not discussed in the review. While shortcomings of citations and references can be fixed with some effort, the edits are unlikely to add currency and relevance to the article.
Major point of critique
A large number of citations are not appropriate. This is a review article for which accuracy of literature is the main objective and this shortcoming must be fixed. The conspicuous absence of certain standard references for candida epidemiology (e.g. Pfaller and Diekema) prompted this reviewer to sample the citations for accuracy. Of the first 10 citations, only 4 were acceptable (1,5,6,8). One was out of date (7), three made no explicit reference to the fact they were supposed to illustrate (2,9,10) and another two were only superficially related to the topic (3 was quoted for “worldwide” while in fact is was specific to Korea, 4 was quoted for candida sp. when in fact is was about C. auris). In the interest of a speedy review that meets the tight timeline of the journal, systematic analysis was discontinued after this initial sampling. However, occasional probing showed that the pattern of poor citations continues throughout the manuscript (e.g. 32 purports to be about C. albicans and non-albicans species, while in fact it is about the somewhat obscure species C. nivariensis; 33 is quoted for lytic enzyme secretion but is in fact about the effect of a combination antifungal therapy on biofilms; 52 is quoted as a reference for “species specificity” of the cell wall when in fact it summarizes the importance of two GPI-anchored cell wall proteins for Slt2 signaling; 67 is quoted for the C. glabrata response to temperature while it is in fact a review of novel glabrata research techniques).
Other points
1. The title. As written, the meaning of the title is not clear. Is C. glabrata adopting (i.e. acquiring) pathogenicity for survival? Against what are the mechanisms resistant? It is likely that the title was supposed to read “Adapting pathogenicity and resistance mechanisms for survival”.
2. Grammar needs work. Some sentences do not make sense because they are missing key elements (e.g. lines 135, 204, 508), others need to be rewritten to clarify the meaning (e.g. line 117). It is of concern that some citations are not complete, suggesting that the automated reference list has been manually edited (e.g. 69 is missing a few words). Citation style in the text is not uniform (lines 430 – 456 list both authors and citation number).
3. C. glabrata and the environment. The global environmental stress response of C. glabrata warrants more extensive discussion (see e.g. Roetzer et al. 2008).
4. Figure 1: The meaning of figure 1 is not clear. Candida glabrata is connected to virulence factors by arrows, but the grouping and the color choice are not explained (do the red arrows means that drug resistance is a special category? How is it different from biofilm formation, and how does iron regulation connect to genetic mutations?)
5. pH controversy. The findings of Yan et al (67; line 305) are discredited without citing the source of contrary evidence.
6. Cryptococcus phagosomal escape (line 322). It is implied that C. neoformans, like C. albicans, escapes the phagosomes by hypha formation. This is not the case – C. neoformans only forms hyphae during the sexual cycle (see e.g. review by Del Poeta et al., 2004).
7. Multidrug resistance introduction (line 379). The entire paragraph does not make sense. While the preceding section argues that MDR is common, the next paragraph states that it is rare and then mentions mutations in a DNA repair gene before summarizing the activity of AMB and echinocandin.
8. Multidrug resistance mechanism (lines 407-418). The discussion bounces between the mechanism of ATP-dependent efflux pumps (410) and the inhibition of ergosterol by azoles (414). As written, the paragraph does not make sense.
9. Listing azole resistances (lines 419-429). This paragraph bundles seemingly random observations on azole resistance in C. glabrata without offering an interpretation.
10. AmphoB. Designating AmB as a “promising” drug is odd – it is licensed since 1959 and well beyond that stage (line 469). A discussion of AmB should include that the drug has severe side effects and is not always “the best therapeutic option” (line 476).
Author Response
Dear Editor,
We appreciate the valuable and constructive comments provided by the REVIEWERS. We have taken them fully into account in our revised manuscript (changes to the text highlighted in yellow). We feel the changes have strengthened the paper, which we hope is now suitable for publication. Our responses to the Reviewer’s comments are in the attached file.
Reviewer 2 Report
The review artilce by Hassan et al, is an extensive and interesting review on Candida glabrata virulence and antifungal resistance mechanisms. While the review article is well written, certain sections needs to be extended to make this a better review. These are as follows:
1) Section 2: A Previous study has shown increased resilience to neutrophil killing in generationally aged C. glabrata cells. The authors need to discuss this.
https://pubmed.ncbi.nlm.nih.gov/28489916/
2)Section 3: Azole resistance: Role of mitochondria in azole resistance is also documented and needs to be discussed in this review.
https://www.ncbi.nlm.nih.gov/pmc/articles/PMC3088236/
3)Section 3:Lines 412-415: Besides SNQ2, CDR1 and CDR2 other genes encoding efflux pumps were also identified that cause drug resistance in C. glabrata. These are listed in the following review and needs to be discussed
https://pubmed.ncbi.nlm.nih.gov/32526921/
4)Section 3: Chromosomal duplication is also documented to cause azole resistance in C. glabrata as described in the following articles:
https://pubmed.ncbi.nlm.nih.gov/19204294/
5)Section 3: Another aspect of antifungal resistance that the authors need to address in the review is the association of replicative aging in C. glabrata drug resistance. The following articles showed increased fluconazole, micafungin and amphotericin B resistance in older generation cells of C. glabrata. The authors should discus these aspects in the review to make it stronger. The articles on aging are as follows:
https://pubmed.ncbi.nlm.nih.gov/28489916/
https://pubmed.ncbi.nlm.nih.gov/33375605/
https://pubmed.ncbi.nlm.nih.gov/29311061/
Author Response
Dear Editor,
We appreciate the valuable and constructive comments provided by the REVIEWERS. We have taken them fully into account in our revised manuscript (changes to the text highlighted in yellow). We feel the changes have strengthened the paper, which we hope is now suitable for publication. Our responses to the Reviewer’s comments are in the attached word document

Round 2
Reviewer 1 Report
Thank you for addressing my comments; I see that my criticisms were taken into consideration and that the manuscript has improved. I have no further suggestions.
Author Response
Thank you for your valuable time in shaping our manuscript.
Reviewer 2 Report
The authors have satisfactorily answered to all my queries. The article has greatly improved.
Author Response

(The authors gave the same response as above.)
